# Homophily Effect in Trauma-Informed Classroom Training for School Personnel

**DOI:** 10.3390/ijerph19127104

**Published:** 2022-06-09

**Authors:** Alexis Zickafoose, Gary Wingenbach, Sana Haddad, Jamie Freeny, Josephine Engels

**Affiliations:** 1Department of Agricultural Leadership, Education and Communications, Texas A&M University, 2116 TAMU, 600 John Kimbrough Blvd., College Station, TX 77843, USA; wingenbach@tamu.edu; 2Mental Health America of Greater Houston, Houston, TX 77098, USA; shaddad@mhahouston.org (S.H.); jfreeny@mhahouston.org (J.F.); jengels@mhahouston.org (J.E.)

**Keywords:** adverse childhood experiences, mental health, train-the-trainer, COVID-19

## Abstract

A national shortage of youth mental health professionals necessitates training others (e.g., school staff) to help youth with behavioral and mental health issues. Professional training in trauma-informed classroom (TIC) practices could increase school staff’s awareness of adverse childhood experiences (ACEs). The purpose was to determine the effect of homophily on participants’ perceptions or knowledge of TIC training. Mental Health America of Greater Houston (MHAGH) offered TIC training from 2019 to 2020 to Texas educators (*N* ≈ 29,900) from nine school districts that experienced significant natural and human-made traumatic events. Proportional stratified random samples were selected based on trainer type (experts vs. peer trainers). Perception was measured with close-ended items on five-point scales. Knowledge was measured with content-specific questions. Independent *t*-tests and two-way ANOVA revealed no significant interaction effects (i.e., trainer and test type) and no differences existed in perception or knowledge by trainer type. TIC training can be equally effective when delivered by homophilous peers (i.e., school staff) and heterophilous experts (i.e., mental health experts). COVID-19 worsened the effects of ACEs and youth mental health issues. High-quality training will increase school staff’s use of TIC practices. MHAGH’s train-the-trainer model helps educators supporting youth affected by ACEs and other life stressors.

## 1. Introduction

Many adolescents have experienced at least one adverse life event (e.g., witnessed parent’s incarceration, domestic violence, neighborhood violence, or racial discrimination) [1,2]. Human-made and natural disasters (such as school shootings, hurricanes, and pandemics) compound adverse life events because of increased psychological challenges, such as familial deaths, severe injuries, and witnessing others being injured [2,3]. After trauma, students are more likely to experience depression, aggression, anxiety, inattention, and sleep issues [2,4,5,6,7,8,9]. The novel coronavirus of 2019 (COVID-19) presented new challenges to students and schools alike. Social isolation and disrupted routines contributed to increased suicide attempts, self-harm, and post-traumatic stress disorder among youth [10,11,12]. Adverse childhood experiences (ACEs) create mental health challenges that affect youth’s abilities to perform in school [13]. Schools are vital short- and long-term recovery centers after natural disasters, assisting in disaster recovery and supporting those affected by trauma [14]. When schools reopened, school personnel needed to rebuild relationships with students to promote safe, welcoming, and predictable environments [15].

Trauma-informed classrooms (TICs) incorporate an understanding of ACEs into classroom cultures to promote safe environments (i.e., physically and psychologically) for student growth [16]. TICs, unlike traditional classroom management techniques (e.g., punishment for student misbehavior for violating classroom rules), recognize that some students’ misbehavior resulting from ACEs is not related to classroom rules or academic content. Rather, TIC techniques (e.g., classroom arrangement to promote quiet/calm zones and classroom safety including safe spaces, empathic teaching/listening models, and zero-difference approach in classroom management) help students cope with the effects of ACEs, which include racial trauma, historical trauma, and natural disasters [17,18,19]. Adequately prepared school personnel can help students cope with these mental health challenges [20,21,22,23,24]. Researchers found that TICs help students manage general behavioral problems and hyperactivity [25,26]. However, [27] found a majority of school personnel nationwide were unaware of TIC strategies.

### Expert vs. Peer Trainers

Homophily is a measure of similarity across characteristics such as beliefs, attitudes, background, and competence [28,29,30]. Much research about teacher homophily found that learners preferred instructors with homophilic characteristics, especially in race, attitude, and background, which increased learners’ satisfaction with instruction [31,32,33]. The authors of [29] suggested balancing homophily and heterophily in teaching/learning situations to encourage the acceptance of new ideas. Previous studies conflict on the effectiveness of homophilous peers and heterophilous experts conducting training. Several found no significant difference between experts’ and trainees’ effectiveness for leading health-related professional development including TIC techniques [22,34,35,36,37,38]. However, other studies showed that expert trainers were rated higher for enthusiasm, credibility, and knowledge [39,40], while peer trainers excelled at interpersonal skills, mental health outcomes, and anxiety and depression skills [36,40,41,42]. Peers tend to be homophilous to each other in background, occupation, and education [28,29,43].

A national shortage of youth mental health professionals necessitates training others to address youth behavioral and mental health issues [11]. If TIC training effectiveness rates are similar when compared by trainer type (mental health expert trainers vs. school “peer” trainers), then the implementation and practice of TIC techniques would be achieved more rapidly, thereby helping more youth affected by ACEs. Proper training in TIC techniques could increase school personnel’s awareness of ACEs. Efficacy of TIC training is vital to ensure participants receive the same quality of information from experts as from expert-trained school peers. Developed following the trauma of Hurricane Harvey, Mental Health America of Greater Houston’s (MHAGH) Emotional Backpack Program (EBP) provided TIC and advanced trauma-informed classroom (T102) training to selected school staff in pre- and ongoing COVID-19 settings.

The purpose of this study was to determine whether homophily affected participants’ knowledge or perceptions of TIC training. Research objectives were to (1) determine whether differences existed in perceptions of TIC and T102 interventions when analyzed by trainer and/or test type; (2) measure TIC pretest knowledge of ACEs and key triggers of trauma; and (3) determine whether differences existed in T102 knowledge of trauma-informed schools and educators when analyzed by trainer type.

## 2. Materials and Methods

This study was conducted as a part of the MHAGH’s Center for School Behavioral Health EBP program, which included similar materials and methods as described herein. A longitudinal trend survey [44] with repeated measures was used for data collection.

The population of interest was all school personnel (*N* ≈ 29,900) from nine Texas independent school districts (Alief, Alvin, Clear Creek, Dickinson, Fort Bend, Goose Creek, Katy, La Porte, and Spring) during the 2019–2020 school year. These school districts experienced significant natural and human-made traumatic events. School personnel received training from expert trainers, who were MHAGH professional development specialists, or from peer trainers (i.e., school personnel within the same school), who were trained originally by MHAGH professional development specialists. Proportionally stratified (by trainer and intervention type) random samples were drawn from the accessible population (*n* = 5367) of TIC and T102 training attendees. Sample sizes were determined using [45] methods for deriving probability samples. Random samples were calculated based on a conservative 50/50 split with a 5% sampling error and a 95% confidence level [45]. Using these parameters, a sample of 356 would adequately represent the population of 5367; we rounded our samples to 400 per training intervention (TIC and T102).

Each intervention included more than 700 participants. We calculated power using G*Power 3.1.9.7 [46]; sensitivity analysis indicated that a sample of 400 was large enough to detect within-subject interactions (i.e., post hoc matched pairs; α = 0.05) as small as *d* = 0.2 with 98% power.

MHAGH provided more than 150 training sessions using EBP interventions, which included TIC and T102 curricula (Figure 1), in Texas school districts from 2019 to 2020. The need for interventions arose because school districts experienced increased student trauma from natural (Hurricane Harvey) and human-made (Santa Fe Texas school shootings) disasters. MHAGH’s in-person workshops were delivered before the novel coronavirus 2019 lockdown; virtual workshops were delivered after March 2020. Three MHAGH mental health experts used a train-the-trainer approach to build school staff capacities for addressing students’ behavioral and mental health development in Texas school districts. Training sessions included experiential activities such as role play simulation, problem solving scenarios, and small group discussions about youth mental health issues. TIC training included how to recognize signs and symptoms of students’ behavioral health issues, effects of trauma on learning, and appropriate responses to students experiencing behavioral and mental health issues. T102 training helped participants learn how to create a trauma-informed classroom. Training sessions were about 90 min each, facilitated by mental health experts or school peer trainers, and included at least two school staff per campus. The training fulfilled the Texas Education Agency requirements for school staff’s professional development in youth mental health and suicide prevention [47].

Each training session included measures of participants’ perceptions and knowledge of TIC and T102 content based on evaluation of learning [48]. Perceptions (on TIC) were measured using multiple retrospective (post-then-pre) questions with Likert-type 5-point scales. Examples of TIC questions included participants’ ratings (5-point scale: low, below average, average, above average, high) of their perceived understanding of trauma’s influence on learning, how to provide a trauma-informed and grief-informed classroom, and ratings of how trauma- and/or grief-informed were their classrooms, schools, and districts. Higher scores indicated positive perceptions about trauma-informed classroom practices. We hypothesized that participants would have more positive perceptions after the intervention. Cronbach’s alphas were 0.89 (before) and 0.92 (after) the TIC intervention; both outcomes were deemed highly reliable. Examples of T102 questions included participants’ ratings (5-point scale: poor, fair, good, very good, excellent) of their perceived knowledge of the meanings of cultural considerations of trauma, historical trauma, racial trauma, implicit bias, and equity. Higher scores indicated more perceived knowledge of these five intervention topics. We hypothesized that participants’ perceived knowledge would be greater after the intervention. Alphas were 0.95 (before) and 0.97 (after) the T102 intervention, also deemed highly reliable.

Two TIC knowledge questions required participants to select the correct choices (*n* = 10) representing traumatic events or ACEs (e.g., sexual abuse, divorce, domestic violence, substance abuse, or natural and human-made disasters), and to correctly identify seven key triggers (e.g., feeling threatened, vulnerable, or ashamed) for students impacted by trauma. Knowledge (T102) was measured by correct choices to two open-ended sentences on trauma-informed schools (consistent, predictable, positive, and safe) and trauma-informed educators (show empathy, foster cultural awareness, and promote equity). Questions on knowledge before or after T102 training were posed as multiple-response (check all correct) items. Testing effect threats were minimized by randomizing the question and response order [49]. Participants’ responses were dichotomously coded (0 = incorrect, 1 = correct). Kuder–Richardson 20 (KR20) reliability coefficients were used to determine the internal consistency of measurements with dichotomous data [44]. KR20 for T102 was 0.71 (before) and 0.64 (after), which were reliable.

School personnel who underwent TIC training had pretest, post-test, 3-month, and end-of-year tests to ascertain knowledge and perception retention. Those who received T102 training sessions had retrospective and end-of-year follow-up surveys only. 

## 3. Results

### 3.1. Participants

More than 5300 school personnel attended MHAGH’s mental health training sessions in 2019 and 2020. TIC training attracted 4262 participants. Of those, 400 were randomly selected for data analysis of TIC techniques. T102 training attracted 1105 participants; 400 were randomly selected for data analysis. The majority were White, female, teachers between the ages of 36 and 55, who attended training sessions led by their peers (Table 1). Participants may not have answered all questions in all tests.

### 3.2. Objective 1

The first objective was to determine whether significant differences existed in perceptions of TIC and T102 interventions when analyzed by trainer and test type. Participants’ perceptions of trauma’s effects on learning were average (*M* = 3.50, *SD* = 1.05) for the MHA trainer pretest but rose above average (*M* = 3.80, *SD* = 1.30) in the post-test. Other tests revealed above average mean (*M* = 3.51–4.50) perceptions for peer trainer and test type (Table 2).

Two-way ANOVA revealed no significant interaction effect between trainer and test type for perceptions of TIC interventions, *F*(1, 321) = 0.47, *p* = 0.494. A simple main effect for test type (pretest vs. post-test, 3-month, and year end) was statistically significant for perceptions of trauma-informed classroom techniques, *F*(3, 321) = 3.70, *p* = 0.012, η_p_^2^
*=* 0.034 (Table 3). Participants’ post-test and year end mean perception scores were significantly higher than were their pretest scores, although the small effect (η_p_^2^
*=* 0.01–0.059) indicates that those gains were of negligible practical or educational difference [50]. Effect sizes were interpreted using [50] standards for partial eta squared (i.e., small = 0.01–0.059, medium = 0.06–0.137, and large = 0.138 or more).

Participants’ perceptions of grief-informed techniques were average in the peer trainer (*M* = 3.50, *SD* = 1.05) and MHA trainer pretests (*M* = 2.83, *SD* = 1.47) but tested above average (*M* = 3.51–4.50) for both trainer type post-tests (Table 4). All means were above average for test type.

Two-way ANOVA revealed no significant interaction effects between trainer and test type on perceptions of grief-informed techniques, *F*(1, 321) = 0.22, *p* = 0.698. A simple main effect for test type (pretest vs. post-test, 3-month, and year-end) was statistically significant for grief-informed classroom techniques, *F*(3, 321) = 4.57, *p* = 0.004, η_p_^2^ = 0.042 (Table 5); a small effect (i.e., negligible practical or educational importance) was noted.

In the T102 intervention, independent *t*-tests were used to calculate whether differences existed in participants’ perceived knowledge of the meanings of various forms of trauma based on trainer type. Pretest perceptions about the meaning of historical trauma were significantly different, *t*(99.54) = −1.99, *p* = 0.049 (small effect, *d* = 0.33 [50]), between peer (*M* = 4.12, *SD* = 0.72) and expert trainers (*M* = 3.85, *SD* = 0.90) (Table 6). No significant differences existed in other tests (i.e., cultural considerations of trauma, racial trauma, implicit bias, and equity). 

### 3.3. Objective 2

The second objective was to measure TIC pretest knowledge of ACEs and key triggers of trauma. Although participants in peer trainer sessions had more knowledge of ACEs (*M* = 7.61, *SD* = 0.93) than did trainees in expert trainer sessions (*M* = 7.30, *SD* = 0.88), the difference was not significant, *t*(381) = −1.76, *p* = 0.079. Likewise, no significant difference in knowledge of the key triggers of trauma were found when analyzed by trainer type, *t*(35.68) = −0.84, *p* = 0.405.

### 3.4. Objective 3

The third objective was to determine whether significant differences existed in T102 knowledge of trauma-informed schools when analyzed by trainer type. Knowledge scores were average (*M* = 2.51–3.50) in all pretests but rose above average (*M* = 3.51–4.50) in all post-tests (Table 7).

A factorial ANOVA revealed statistically significant differences by test type, *F*(1, 370) = 28.82, *p* < 0.001; there were no significant interaction effects for knowledge of trauma-informed schools (Table 8). Post-test knowledge (*M* = 3.57, *SD* = 0.72) was significantly greater than pretest knowledge (*M* = 3.06, *SD* = 1.04). A medium effect size for test type (η_p_^2^ = 0.073) may help explain the non-significant interaction effect.

Knowledge of trauma-informed educator’s roles was average (*M* = 2.51–3.50) for all pretest/post-test scenarios when analyzed by trainer and test type (Table 9). 

A significant difference, *F*(1, 364) = 5.10, *p* = 0.024, η*_p_*^2^ = 0.014, existed for participants’ knowledge of trauma-informed educator’s roles when analyzed by test type (Table 10), although no interaction effect was found, *F*(1, 364) = 1.13, *p* = 0.288. Pretest knowledge (*M* = 2.82, *SD* = 0.50) was significantly lower than post-test knowledge (*M* = 2.84, *SD* = 0.47).

## 4. Discussion

### 4.1. Perceptions

The findings of no differences in perception based on trainer type echo those of [34]. Our findings are consistent also with studies in health-related professional development [22,35,36,37,38]. This finding is important because it further reinforces the efficacy of the train-the-trainer model. Concerning the homophily effect, peer trainers’ homophilous education and occupation backgrounds did not significantly affect school personnel’s perceptions of trauma-informed classroom techniques. This finding contradicts the importance placed on homophily in previous studies [29,30]. MHAGH can reach more school personnel and, therefore, foster more TIC practices by continuing its train-the-trainer model. It is concerning that the majority of the perception questions produced no differences based on test type. Significant differences in perceptions of trauma’s effects on learning between pretest/post-test and pretest/year-end tests suggest that following TIC training, perceptions increased and remained so for considerable time. For perception of grief-informed techniques, the large effect size between all tests suggests a more long-term change in perception after TIC training. Ideally, there would be a difference based on test type pointing to elevated perceptions of TIC practices. Because this is not evident, TIC training may need to provide participants with more clarification and practice in using TIC practices during training sessions.

### 4.2. Knowledge

The lack of significant differences in prior knowledge of TIC practices could mean that trainees had similar experiences and information about TIC practices before their training. Considering when these training sessions took place, the expert-led sessions always preceded peer-led sessions in the same school district. Therefore, if trainees in peer-led training sessions already knew about TIC practices before their training sessions, then their peers may have informally shared the knowledge gained from their expert training sessions. On one hand, the lack of significant differences may present a threat to internal validity through testing effect. On the other hand, peer trainers’ potential excitement to share knowledge gains from the expert-led training sessions bolsters MHAGH’s use of the train-the-trainer model for rapid dissemination of TIC and T102 curricula. 

The lack of statistical significance in knowledge of TIC techniques for T102 participants mirrors the findings of [34], who found no difference in the short-term knowledge increase of TIC techniques when analyzed by trainer type (experts vs. peer trainers). Knowledge gain was similar regardless of trainer type. Therefore, knowledge gains through T102 training could lead to “increased capacity” in supporting youth affected by ACEs. Since there are no differences based on trainer type, the importance of homophily in learning was not a significant factor in this case [28]. While there were negligible differences by test type, we would expect greater differences between pre- and post-tests. The lack in differences signals that modifications may be needed during training, such as clarifying the characteristics of trauma-informed schools and the roles of trauma-informed educators, as well as allowing participants to practice TIC techniques, with feedback, during training sessions.

### 4.3. Limitations

This study has some limitations. While a stratified random sample was used, it was drawn from an expert sample of registrants (i.e., interested educators) in TIC and T102 training sessions. A truly random sample of K-12 school personnel would provide more accurate insights about the effects of TIC training.

The study was limited by the lack of adequate time to provide multiple practice elements during training sessions. However, the literature remains mixed on which training methods lead to a successful transfer of training [51,52]. In both TIC and T102 trainings, trainers provided participants with opportunities to discuss key takeaways from instructional videos and reflection time at the end of each training, thus promoting deeper impact on practice.

Another limitation exists in the measurements of knowledge. The T102 knowledge portion required four correct situations be selected as correct, rather than multiple choice questions with a single correct answer per question. We did not count partially correct answers; therefore, true knowledge gains could not be assessed. Differences in test construction and administration may have affected the data set. In future iterations, these issues will be corrected with long-term use of standardized tests that increase instrument fidelity.

Finally, the lack of a control group limits the reported data. MHAGH seeks to conduct experimental trials of TIC and other curricula to accurately measure the effects of training for school personnel associated with youth behavioral and mental health development issues. Increased efforts to establish experimental or quasi-experimental conditions for testing TIC/T102 curricula will heighten research fidelity and rigor.

## 5. Conclusions

TIC training for school personnel is vital for supporting students and others who cope with ACEs and other forms of trauma. Our findings support the implementation of the train-the-trainer model to create awareness and increase knowledge of TIC practices. MHAGH should consider amending its training program to clarify TIC skills, roles for educators, and characteristics of trauma-informed schools. Future research should focus on the impacts of trainings on students and classroom environments. Communities want and need healthy, productive, and engaged youth. Positive youth development is part of every educational system when TIC practices are understood and used by school personnel. Communities of health are created when TIC practices are incorporated into daily living, in and outside the school system.

COVID-19 and other life stressors have further aggravated ACEs and some youth’s mental health issues. The national shortage of youth mental health professionals will not be alleviated soon, yet children’s mental health was affected negatively by the pandemic [53]. There is increased urgency for high-quality training to strengthen school staff’s implementation of TIC practices in all classrooms. 

## Figures and Tables

**Figure 1 ijerph-19-07104-f001:**
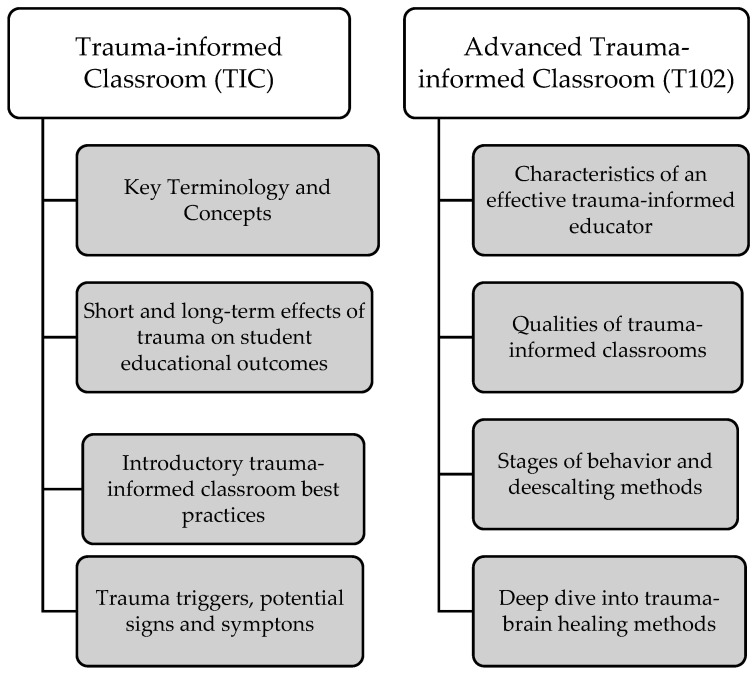
TIC and T102 training components.

**Table 1 ijerph-19-07104-t001:** Frequencies for nominal and ordinal independent and dependent variables (*n* = 800).

Variables (Categories)		TIC		T102	
		*f*	%	*f*	%
Trainer					
	Peer trainer	369	92.3	281	70.3
	Expert trainer	31	7.8	119	29.8
Tests					
	Pretest	226	56.5	192	48.0
	Post-test	143	35.8	180	45.0
	3-month	21	5.3		
	Year end	10	2.5	28	7.0
Title					
	Teacher	171	75.7	143	45.0
	Counselor	10	4.4	86	27.0
	Another title	28	12.4	44	13.8
	Paraprofessional	15	6.6	36	11.4
	Administrator	2	0.9	9	2.8
Sex					
	Female	113	78.5	309	89.6
	Male	31	21.5	36	10.4
Race/Ethnicity					
	White	78	55.7	147	47.9
	Hispanic	28	20.0	77	25.1
	Black/African American	21	15.0	57	19.2
	Another	13	9.3	26	8.8
Age					
	36–55	62	44.0	171	60.9
	18–35	61	43.3	73	26.0
	56+	18	12.7	24	8.5
	Another			13	4.6

*Note*. Frequencies may not equal 100% because of missing data.

**Table 2 ijerph-19-07104-t002:** Perceptions of trauma’s effects on learning (*n* = 321).

Variables	Tests	*n*	*M*	*SD*
Peer Trainer				
	Pretest	142	3.54	0.89
	Post-test	138	4.41	0.67
	3-month	21	4.10	0.63
	Year end	10	4.30	0.68
MHA Trainer				
	Pretest	6	3.50	1.05
	Post-test	5	3.80	1.30
Total				
	Pretest	148	3.54	0.89
	Post-test	143	4.38	0.70
	3-month	21	4.10	0.63
	Year end	10	4.30	0.68

*Note*. Measured on a 5-point Likert scale with 1 = low and 5 = high.

**Table 3 ijerph-19-07104-t003:** Two-way ANOVA for trauma-informed techniques by trainer and test type (*n* = 321).

Source	SS	*df*	MS	*F*	*p*	η_p_^2^
Corrected Model	71.43 ^a^	5	14.29	20.45	<0.001 *	0.244
Intercept	592.83	1	592.83	848.48	<0.001 *	0.729
Trainer	0.98	1	0.98	1.41	0.237	0.004
Test	7.76	3	2.59	3.70	0.012*	0.034
Trainer * Test	0.33	1	0.33	0.47	0.494	0.001
Error	220.79	322	0.70			
Total	292.21	321				

*Note*. ^a^ *R*^2^ = 0.24, adjusted *R*^2^ = 0.23, * *p* < 0.05.

**Table 4 ijerph-19-07104-t004:** Descriptive statistics for perceptions of grief-informed techniques (*n* = 321).

Variables	Tests	*n*	*M*	*SD*
Peer Trainer				
	Pretest	142	3.19	0.97
	Post-test	138	4.21	0.72
	3-month	21	3.95	0.74
	Year end	10	4.00	0.82
MHA Trainer				
	Pretest	6	2.83	1.47
	Post-test	5	3.60	1.52
Total				
	Pretest	148	3.18	0.99
	Post-test	143	4.19	0.76
	3-month	21	3.95	0.74
	Year end	10	4.00	0.82

*Note.* Measured on a 5-point Likert scale with 1 = low and 5 = high.

**Table 5 ijerph-19-07104-t005:** Two-way ANOVA for grief-informed techniques by trainer and test type (*n* = 321).

Source	SS	*df*	MS	*F*	*p*	η_p_^2^
Corrected Model	79.62 ^a^	5	15.92	20.99	<0.001 *	0.249
Intercept	560.71	1	560.71	739.02	<0.001 *	0.700
Trainer	2.54	1	2.45	3.24	0.073	0.010
Test	10.40	3	3.47	4.57	0.004 *	0.042
Trainer * Test	0.17	1	0.17	0.22	0.638	0.001
Error	239.76	316	0.76			
Total	319.38	321				

*Note*. ^a^*R^2^* = 0.25, adjusted *R^2^* = 0.24, * *p* < 0.05.

**Table 6 ijerph-19-07104-t006:** Independent t-test for T102 pretest perception of historical trauma’s meaning.

Treatment Group	*n*	*M*	*SD*	*t*	*p*	*d*
Peer trainers	130	4.12	0.72	−1.99	0.049	0.33
Expert trainers	62	3.85	0.90			

*Note*. 5-point Likert-type scale (1 = poor to 5 = excellent).

**Table 7 ijerph-19-07104-t007:** Descriptive statistics for knowledge of trauma-informed schools (*n* = 371).

Variables	Tests	*n*	*M*	*SD*
MHA Trainer				
	Pretest	61	3.10	1.06
	Post-test	57	3.70	0.60
Peer Trainer				
	Pretest	130	3.04	1.03
	Post-test	123	3.51	0.77
Total				
	Pretest	191	3.06	1.04
	Post-test	180	3.57	0.72

*Note.* The number of correct selections for this question was 4.

**Table 8 ijerph-19-07104-t008:** Two-way ANOVA for knowledge of trauma-informed schools (*n* = 371).

Source	SS	*df*	MS	*F*	*p*	η*_p_^2^*
Corrected Model	26.09 ^a^	3	8.70	10.75	<0.001 *	0.081
Intercept	3582.08	1	3582.08	4428.14	<0.001 *	0.923
Trainer	1.25	1	1.25	1.55	0.215	0.004
Test	23.32	1	23.32	28.82	<0.001 *	0.073
Trainer * Test	0.34	1	0.34	0.42	0.519	0.001
Error	296.88	367	0.81			
Total	322.97	371				

*Note*. ^a^*R^2^* = 0.08, adjusted *R^2^* = 0.07, * *p* < 0.05.

**Table 9 ijerph-19-07104-t009:** Descriptive statistics for knowledge of trauma-informed educators’ roles (*n* = 365).

Variables	Tests	*n*	*M*	*SD*
MHA Trainer				
	Pretest	61	2.79	0.56
	Post-test	57	2.79	0.53
Peer Trainer				
	Pretest	127	2.85	0.43
	Post-test	120	2.91	0.37
Total				
	Pretest	188	2.82	0.50
	Post-test	177	2.84	0.47

*Note.* The number of correct selections for this question was 4.

**Table 10 ijerph-19-07104-t010:** Two-way ANOVA for knowledge of trauma-informed educators’ roles (*n* = 365).

Source	SS	*df*	MS	*F*	*p*	η*_p_^2^*
Corrected Model	1.52 ^a^	3	0.51	1.90	0.130	0.016
Intercept	2523.25	1	2523.25	9460.80	<0.001 *	0.963
Trainer	0.15	1	0.15	0.56	0.454	0.002
Test	1.36	1	1.36	5.10	0.024 *	0.014
Trainer * Test	0.30	1	0.30	1.13	0.288	0.003
Error	96.28	361	0.27			
Total	97.80	364				

*Note*. ^a^*R^2^* = 0.02, adjusted *R^2^* = 0.01, * *p* < 0.05.

## Data Availability

Not applicable.

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
