# Peer review of "Homophily Effect in Trauma-Informed Classroom Training for School Personnel"

_ijerph, 2022, doi:10.3390/ijerph19127104_

Round 1

Reviewer 1 Report

Thank you for the opportunity to review your manuscript on TIC in schools. It is well written and an excellent contribution to the literature in this area.

This section is confusing - homophily and heterophily in teaching/learning situations to encourage acceptance of new ideas. Previous studies conflict on the effectiveness of homophilous peers and heterophilous experts conducting training. Several found no significant difference between experts’ and trainees’ effectiveness

are trainees always peers? Are peers always homophilous? That seems unlikely to me as you explain homophiliy is about beliefs, attitudes and background not expertise. This section is important to explain homophiliy and needs revising. The school peer trainers are not described. How do they differ from experts? I learned in the results section that the peer trainers were teachers, so peers of teachers not peers of students. This still does not address homophiliy.

The methods and results are clearly described and explained.

discussion/ conclusion - line251. What does it mean to unobscure TIC skills?

There is no reference to homophiliy in the discussion or conclusion. Is it even relevant? The literature section might be better focused on train the trainer literature and training transfer rather than homophiliy.

I think the limitations should include that skills were not practiced in the training. The training transfer literature is clear that training alone has minimal impact on practice. This is more of a limitation in the program than the research. For example in the materials and methods - T102 training helped participants learn how to create a trauma-informed classroom (in a 90 minute session). The section describes some of the training elements but very briefly. Can a couple more sentences be added about practice elements or lack of?

the suggestion about future research including impact on students is important as uti at Ely this is the test of the program.

Reviewer 2 Report

I have read the manuscript entitled "Homophily Effect in Trauma-informed Classroom Training for School Personnel" with interest. This study compared scores of perceptions and knowledges between TIC training and T102 training participants. The study addresses an important but currently under-researched topic in public mental health. This manuscript is very suitable to the special issue. Although the study has certain potential, I feel that important information is missing, especially in the Materials and Methods and Results sections. I have several suggestions to improve readability and clarity.

1) I suggest adding a definition of expert (probably to the second paragraph of the Materials and Methods section). One of the most important variables of the study is whether the trainer is expert or not. It is important for readers to understand how the expert has been defined in the study. 

2) I suggest adding a table or a figure that describes the differences of interventions between two conditions namely TIC and T102 (probably to the fourth paragraph of the Materials and Methods section). One of the most important variables of the study is whether the training is TIC or T102. The contents of these training programs are described only in sentences. I believe that a table or a figure will make the differences clear.

3) I can not understand what measures of participants’ perceptions and knowledge of TIC and T102 (probably to the fifth paragraph of the Materials and Methods section) assess. Sample items and meanings of scores (ie, higher scores indicate what) seem necessary.

4) I suggest adding means and standard deviations of the main outcomes to the Results section. Table 2, Table 3, Table 4, and Table 5 report results of ANOVAs. However, I believe that mean scores of main outcomes (perceptions and knowledges) by trainer and training with SDs should be reported.

Round 2

Reviewer 2 Report

Thank you very much for revising the manuscript according to my comments. I have confirmed that my all comments had been addressed. I hope that your manuscript will be published and will stimulate this important but currently under-researched topic. I have found a few points that authors might want to address. I am not good at statistics. If my comments do not make sense, I appreciate if you would ignore my comments.  

1) Page 6, Table 3, df of Test. The degree of freedom for Test is 3. I am wondering if “1” might be correct. The degree of freedom of the interaction is 1.

2) It might be better to refer to the guideline for interpreting effect sizes.

Thank you.
